# BIDDING FOR INFLUENCE: AUCTION-DRIVEN DIFFUSION IMAGE GENERATION

## ABSTRACT

Motivated by online auctions for banner ads, we propose auctions that fractionally allocate the creation of a banner to bidders according to their preferences. Our mechanism elicits bids and textual prompts from the advertisers, and composes them into a score function that drives a reverse diffusion process that generates the banner. Then, it implements Monte Carlo sampling to calculate approximate VCG-based payments to incentivize high-welfare images. Extensive experiments on a diverse 20-prompt dataset with up to 3 agents demonstrate key economic properties. Our mechanism achieves: (1) bid monotonicity; (2) efficiency improvement of up to 20.7% higher welfare than a single-winner VCG baseline; and (3) approximate incentive compatibility, with average regret as low as 7% when deviating from truthful bidding. These benefits are achieved while preserving high image quality. Our study establishes a principled and scalable bridge between auction theory and controllable image diffusion, laying a foundation for economically aligned, multi-stakeholder image generation in advertising and beyond.

## 1 INTRODUCTION

Online advertising auctions, which power sponsored search and banner ads, are a cornerstone of the digital economy. A fundamental limitation of these auctions, however, is that they allocate *discrete*, indivisible resources—such as a single ad slot on a webpage. This structure inherently leads to winner-take-all outcomes, where one advertiser's win precludes all others, preventing any form of shared value creation from a single advertising opportunity.

Generative AI marks a paradigm shift from this limitation. The "good" being auctioned is no longer a fixed, discrete slot but a *continuous and malleable outcome*, such as a piece of text or an image. This transformation opens the door for multi-winner auctions where several agents can be represented in

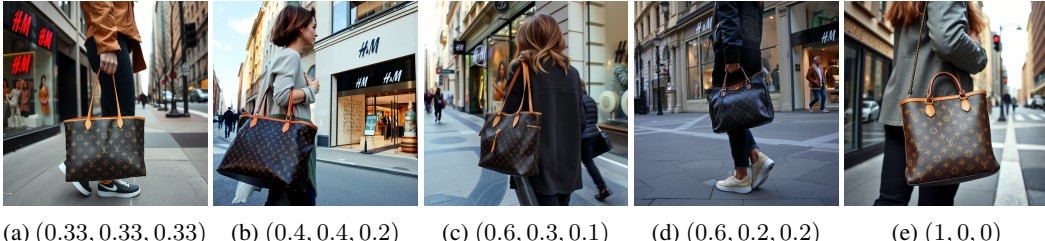

(a) $(0.33, 0.33, 0.33)$    (b) $(0.4, 0.4, 0.2)$    (c) $(0.6, 0.3, 0.1)$    (d) $(0.6, 0.2, 0.2)$    (e) $(1, 0, 0)$

Figure 1: Visual examples of our generative auction mechanism. This scenario simulates an auction between three agents (Louis Vuitton, H&M, and Nike) bidding for influence in an image generated from the base prompt "A person walking down a city street." The terms $(b_1, b_2, b_3)$ represent the normalized bids for each respective agent. From left to right, the images show the visual output as the bid distribution shifts from balanced, where all agents are equally represented, to scenarios with one or two dominant agents, and finally to a winner-take-all case. This fine-grained, bid-dependent control is achieved by our mechanism aggregating agent preferences directly within the diffusion model's score function, creating a scalable auction for a continuous, visual outcome.

a single, dynamically generated output. Consider a user querying a Large Language Model (LLM) with "*What should I do in Boston?*". The response could be:

> Boston is a great city for Italian food... Try {`<Italian Restaurant Ad 1>`} in the North End, then stop by {`<Local Shop Ad 2>`} for souvenirs.

In this generative context, multiple advertisers win exposure from a single query, each with a different level of prominence. While this concept has been explored for LLMs (Dütting et al., 2024; Hajiaghayi et al., 2024; Feizi et al., 2024), applying it to the high-dimensional, visual domain of image synthesis presents unique challenges, such as aggregating entangled preferences in a continuous pixel space and ensuring visual coherence during composition

This paper introduces the first auction mechanism specifically for diffusion-based image generation. While significant research has shown that diffusion models can be technically steered to combine multiple concepts into a single coherent image (Ho & Salimans, 2022), no principled framework exists to allow multiple agents to bid for and share influence over the final visual content. This gap motivates our central research question: How can we design an effective, generative image auction that enables multiple agents to bid for influencing the content of the generated image?

To solve this, we propose an auction mechanism that translates bids into a dynamic composition of score functions within the diffusion model's reverse process. Our main contributions include:

- **New Class of Generative Auctions:** We introduce the problem of *generative image auctions* and propose the first complete, scalable mechanism that allows multiple agents to bid for fractional influence over a shared, continuous visual outcome.
- **Allocation and Pricing Framework:** We design a mechanism that translates bids into visual influence via a score composition technique inspired by classifier-free guidance. This allocation rule is paired with a payment rule that approximates VCG principles using Monte Carlo sampling to ensure economic robustness.
- **Comprehensive Empirical Validation:** We demonstrate through extensive multi-agent experiments on a diverse dataset that our mechanism achieves key economic properties: significant welfare improvement (up to 20.7%) over single-winner baselines, approximate incentive compatibility, and bid monotonicity, all without degrading image quality.

We believe this work lays a foundational framework for scalable, controllable, and economically-efficient multi-stakeholder image generation.

## 2 RELATED WORK

### 2.1 AUCTION THEORY AND MECHANISM DESIGN

Our work is fundamentally inspired by the principles of auction theory, with a particular focus on the Vickrey-Clarke-Groves (VCG) mechanism (Vickrey, 1961; Clarke, 1971; Groves, 1973). In the classic single-object auction setting, VCG stands out for its elegant properties: it allocates an item to the bidder with the highest valuation and sets the payment based on the opportunity cost imposed on others. This guarantees both social welfare maximization and incentive compatibility, ensuring that an agent's optimal strategy is to bid truthfully. Applying the VCG mechanism to the domain of AI-generated content, where the outcome is continuous and shared, presents new conceptual problems and opportunities.

### 2.2 GENERATIVE AUCTIONS FOR LANGUAGE MODELS

Recent research has begun to explore the intersection of auctions and LLMs (Feizi et al., 2024). A central challenge is aggregating preferences from multiple agents. To this end, Dütting et al. (2024) introduced token auctions where bids influence the probability of the next generated token. Others have focused on contextually relevant ad placement within LLM outputs using retrieval-augmented generation (RAG) (Hajiaghayi et al., 2024) or in summaries (Dubey et al., 2024). While these works establish a foundation for generative auctions in text, they leave a notable gap concerning multi-agent preference aggregation for image generation, which our work aims to address.

## 2.3 Controllable Image Generation with Diffusion Models

To realize a generative auction in the visual domain, we build upon the technical foundation of diffusion models, which have achieved state-of-the-art results in synthesizing diverse, high-fidelity images (Ho et al., 2020; Song & Ermon, 2019; Rombach et al., 2022; Dhariwal & Nichol, 2021). These models learn to generate data by reversing a gradual noising process, progressively removing noise from a random tensor to produce a clean image sample (Sohl-Dickstein et al., 2015; Ho et al., 2020; Song et al., 2021). Recent advances, such as Rectified Flow, have further improved generation speed and quality (Liu et al., 2022; Esser et al., 2024).

A crucial aspect of this technology for our purposes is its controllability. Generation can be steered toward desired attributes using guidance techniques. Classifier-free guidance (CFG) has become a prominent method, as it allows for strong conditional control without a separate classifier model (Ho & Salimans, 2022). Its effectiveness in high-fidelity text-to-image synthesis has been well-demonstrated (Saharia et al., 2022; Nichol et al., 2022). Our work leverages this technical capability not just for creative control, but as a primitive upon which we build a formal auction mechanism for integrating multiple agent preferences into a single generated image.

## 3 Methodology

Our goal is to design an auction mechanism capable of generating a single, shared image that effectively aggregates the preferences of multiple agents. The mechanism must define how agents express their preferences, how these preferences are combined to generate an outcome (the allocation rule), and what agents pay (the payment rule). We design our mechanism to satisfy key economic properties: (a) bid monotonicity, where influence increases with bids; (b) welfare improvement over single-winner baselines; and (c) approximate incentive compatibility, where truthful bidding is incentivized, all while maintaining high image quality.

### 3.1 Preliminaries

We first review the foundational concepts from mechanism design and diffusion models that our work builds upon.

#### 3.1.1 VCG Auctions and Social Welfare

In auction theory, a mechanism's goal is often to maximize *social welfare*, defined as the sum of all agents' valuations for a given outcome. For a set of agents $N$ and a chosen outcome (in our case, an image $I$), the social welfare is

$$W(I) = \sum_{i \in N} v_i(I), \tag{1}$$

where $v_i(I)$ is the value agent $i$ derives from image $I$. The Vickrey-Clarke-Groves (VCG) mechanism is a celebrated result that achieves maximum social welfare while also being *incentive compatible* (truthful). It consists of two parts:

1. **Allocation Rule:** Choose the outcome $I^*$ that maximizes the declared social welfare.
2. **Payment Rule:** Charge each agent $i$ for the *externality* they impose on others, calculated as the welfare the other agents would have gotten if $i$ had not participated, minus the welfare they get now that $i$ is present.

#### 3.1.2 Score-Based Diffusion and Guidance

Denoising diffusion models learn to generate data by reversing a noising process. This is often achieved by learning a score function, $s_t(x|c) = \nabla_{x_t} \log p_t(x_t|c)$, which guides the generation of an image $x$ at time $t$ based on a condition $c$ (e.g., a text prompt). A key technique for controlling this process is Classifier-Free Guidance (CFG) (Ho & Salimans, 2022). CFG sharpens the generation by steering it further toward the conditional score and away from an unconditional score $s_t(x|\emptyset)$. The guided score $\hat{s}_t$ is typically formulated as:

$$\hat{s}_t(x|c) = (1 - \omega)s_t(x|\emptyset) + (\omega)s_t(x|c) \tag{2}$$

where $\omega$ denotes the guidance scale. Increasing $\omega$ amplifies the conditioning signal.

## 3.2 The Generative Image Auction Mechanism

We now formalize the components of our generative image auction.

1. **Base Prompt** ($c$)**:** A text prompt describing the base content of the image.
2. **Agents** ($N = \{1, \ldots, n\}$)**:** Each agent $i$ wishes to influence the final image.
3. **Agent Inputs:** Each agent $i$ has a public text prompt $c_i$ describing their desired content and submits a private bid $b_i \in \mathbb{R}_{\geq 0}$ to the auction.
4. **Agent Valuation:** Each agent has a private value multiplier $m_i$ and derives value from an image $I$ according to the function $v_i(I) = m_i \cdot \alpha_i(I)$, where $\alpha_i(I)$ is function measuring the alignment between image $I$ and prompt $c_i$.

Given these components, our VCG-inspired mechanism implements an **Allocation Rule** (Section 3.3) to generate a high-welfare image and a **Payment Rule** (Section 3.4) to charge the agents.

## 3.3 The Allocation Rule: From Bids to Images via Score Composition

The core of our mechanism is a score composition technique, inspired by classifier-free guidance, that translates agent bids into fine-grained control over the diffusion process. Instead of amplifying a single condition against an unconditional score, it performs a bid-weighted interpolation between two different conditional states, a cooperative joint-prompt state and a dominant-agent state.

For clarity, we first present the technique for 2 agents, assuming $b_1 \geq b_2$ without loss of generality. We define normalized bids $b_1^{(1)} = \frac{b_1}{b_1 + b_2}$ and a weight $w^{(1)} = 2b_1^{(1)} - 1$. The composed score $s_t^{(1,2)}(x)$ is:

$$s_t^{(1,2)}(x) = (1 - w^{(1)})s_t(x|c, c_{1,2}) + (w^{(1)})s_t(x|c, c_1) \tag{3}$$

where $c_{1,2} = \{c_1, c_2\}$ is the joint prompt. When bids are equal ($b_1^{(1)} = 0.5$), then $w^{(1)} = 0$ and the score relies solely on the cooperative joint prompt. When Agent 1 is fully dominant ($b_1^{(1)} = 1$), $w^{(1)} = 1$, and the score collapses to the dominant agent's prompt.

We generalize the score composition technique to $n$ agents recursively, with the full derivation in Appendix B. The score for $n$ agents is defined as:

$$s_{1,n}^*(x) = (1 - w^{(1)})\, s(x|c, c_{1,n}) + (w^{(1)})\, s_{1,n-1}^*(x) \tag{4}$$

where $s_{1,n-1}^*(x)$ is the composed score for the $n - 1$ most dominant agents.

Finally, to select an outcome, we follow the VCG allocation principle. We use the composed score to run the reverse diffusion process $k$ times, generating a set of candidate images. We then select the single image $\hat{I}$ that maximizes the total welfare, as calculated using the agents' submitted bids.

## 3.4 The Payment Rule: VCG-Inspired Pricing

Following the VCG principle described in Section 3.1, our payment rule charges each agent for the externality they impose on other participants. Because we cannot analytically compute the exact maximum welfare, we approximate the VCG payments using Monte Carlo sampling.

To compute the price for agent $i$, we first calculate the realized welfare of all other agents from the chosen image $\hat{I}$. We then re-run the entire allocation process (Section 3.3) in a counterfactual world without agent $i$ to find the image $\hat{I}_{-i}$ that would have been chosen in their absence. The payment for agent $i$ is the difference between the welfare others would have received without them and the welfare they actually received with them:

$$p_i = \left(\sum_{j \neq i} v_j(\hat{I}_{-i})\right) - \left(\sum_{j \neq i} v_j(\hat{I})\right) \tag{5}$$

This ensures that agents are charged based on the social cost of their participation.

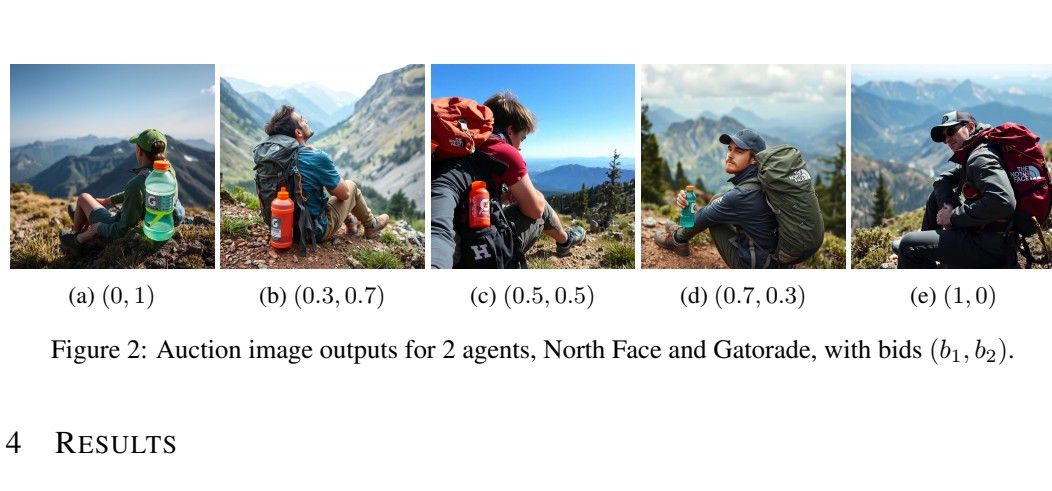

| (a) $(0,1)$ | (b) $(0.3, 0.7)$ | (c) $(0.5, 0.5)$ | (d) $(0.7, 0.3)$ | (e) $(1, 0)$ |

Figure 2: Auction image outputs for 2 agents, North Face and Gatorade, with bids $(b_1, b_2)$.

## 4 RESULTS

### 4.1 EXPERIMENTAL SETUP

To evaluate how effectively our auction blends multiple advertiser concepts within a single generated image, we construct a controlled test set of 20 base prompts. We first explore the 2-agent setting. Each base prompt is paired with two advertiser prompts, along with a joint prompt that combines both agents' preferences. For example, the following prompts correspond with Figure 2:

- **Base:** "A hiker resting on a mountain trail"
- **Base + Agent 1:** "A hiker resting on a mountain trail with North Face backpack"
- **Base + Agent 2:** "A hiker resting on a mountain trail with Gatorade bottle"
- **Base + Joint:** "A hiker resting on a mountain trail with North Face backpack and Gatorade bottle"

We then study the 3-agent setting, introducing an additional agent for every base prompt, to demonstrate the robustness of our mechanism for multiple agents. Appendix A includes full prompts. This setup allows us to systematically vary each agent's bid weight while holding prompts constant, allowing a full range of score compositions to generate a set of final images.

For the 2-agent case, we evaluate seven combinations that sweep the full spectrum of influence: $(0.0, 1.0), (0.1, 0.9), (0.3, 0.7), (0.5, 0.5), (0.7, 0.3), (0.9, 0.1)$, and $(1.0, 0.0)$. For the 3-agent case, we evaluate five representative bidding scenarios to probe key dynamics: an all-equal distribution $(0.33, 0.33, 0.33)$, a ranked-bidding scenario $(0.6, 0.3, 0.1)$, a winner-take-all case $(1.0, 0.0, 0.0)$, and scenarios with one $(0.6, 0.2, 0.2)$ or two $(0.4, 0.4, 0.2)$ dominant agents. Figure 1 displays example outputs for 3 agents.

We conduct Monte Carlo sampling, generating $k$ images per bid combination where $k$ denotes the sample size. All images are generated using the `FLUX.1-schnell` diffusion model with 5 inference steps and a guidance scale of 10. We use CLIP (Radford et al., 2021) to compute alignment $\alpha_i = \text{CLIP}(I, c_i)$ as the cosine similarity between the CLIP image embedding and text embedding, which lie in a shared embedding space.

### 4.2 BID MONOTONICITY: PROMPT ALIGNMENT INCREASES WITH BID

One auction trait that we aim to show is bid monotonicity, in which agents who bid more should receive higher valuations. Then, an advertiser's prompt should be more prominent in the generated image as their bid increases. To evaluate this, we increase agent 1's normalized bid values, which decreases agent 2's normalized bids, across all five prompts. For each bid pair, we generate $k = 20$ images using our score composition method and compute the average prompt alignment scores to test whether higher bids yield stronger semantic alignment.

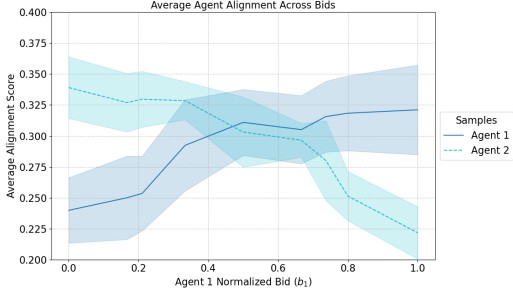

Figure 3: Auction mechanism satisfies bid monotonicity.

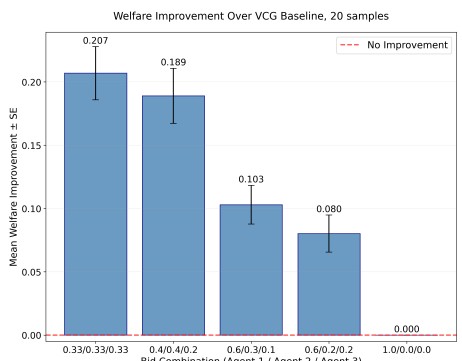

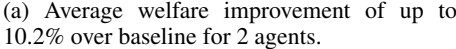

(a) Average welfare improvement of up to 10.2% over baseline for 2 agents.

(b) Average welfare improvement of up to 20.7% over baseline for 3 agents.

Figure 4: Our diffusion image auction significantly improves welfare over the single-winner VCG auction baseline, averaged across 20 prompts. Each setting had $k = 20$ Monte Carlo samples.

**Results.** Figure 3 demonstrates that as agent 1's bid increases, its prompt alignment improves monotonically from 0.25 to 0.32, reflecting a higher realized valuation. Conversely, as agent 2's bid decreases, its prompt alignment degrades monotonically from 0.34 to 0.22, indicating reduced influence over the generated content. These results validate that as advertisers bid more, they exert more influence on the generated image, as visualized in Figure 2.

### 4.3 EFFICIENCY: IMPROVED WELFARE COMPARED TO SINGLE-WINNER BASELINE

Unlike single-winner auctions, our mechanism enables multiple agents to be represented simultaneously in the same generated output. A key goal is to show that this compositional capability leads to higher total welfare.

**Single-Winner VCG Baseline.** To evaluate this, we compare our method against a single-winner VCG auction baseline that generates images using only one agent's prompt. In this baseline, $2k$ images are sampled, $k$ images using each agent's prompt appended to the base prompt. The final output is the image that optimizes overall welfare. This baseline serves as a direct comparison without agent preference compositionality.

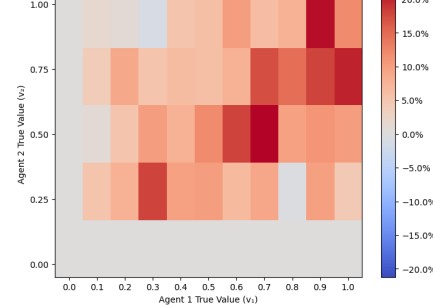

Figure 5: Welfare improvement is most significant when agents bid similar values.

**Results.** Our diffusion image auction consistently outperforms the single-winner VCG baseline in optimizing welfare. For the 2-agent setting, Figure 6a shows that our mechanism improves average welfare by up to 10.2% over the baseline, which occurs when both agents' normalized bids are equal to 0.5. Figure 5 demonstrates up to 21% welfare improvement for a specific bid combination on an individual prompt in the 2-agent setting ("A car driving along a scenic coastal highway" in Table 3).

For the 3-agent setting, Figure 4b shows that our mechanism improves average welfare by up to 20.7% over the baseline, which occurs when all 3 agents' normalized bids are equal to 0.33. Individual prompts demonstrate up to 40% welfare improvement for a specific bid combination.

To systematically characterize welfare trends in the 2-agent setting, we set $k = 50$, varying Agent 1's bid from 0.0 to 1.0 in increments of 0.1 and Agent 2's bid from 0.0 to 1.0 in increments of 0.25. As shown in Figure 5, the welfare improvement over the baseline is highest when agents bid similar values (along the $b_1 = b_2$ diagonal). This peak occurs because balanced bids activate the cooperative potential of our score composition, guiding the model to find a harmonious blend. In doing so, our

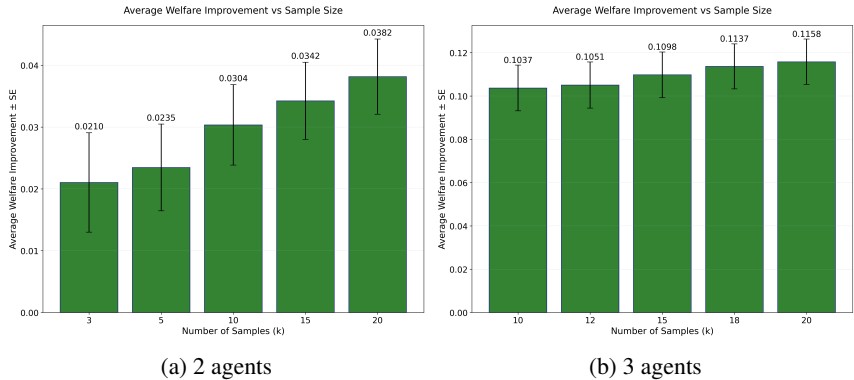

(a) 2 agents                    (b) 3 agents

Figure 6: Increasing Monte Carlo sample size monotonically increases welfare improvement over single-winner baseline.

mechanism creates a new, high-welfare outcome through cooperation that is entirely inaccessible to the single-winner system, which is limited to satisfying only one agent at a time.

Furthermore, the efficiency of our mechanism is directly enhanced by the sampling budget. As illustrated in Figure 6, a larger number of Monte Carlo samples allows the mechanism to more effectively identify high-welfare outcomes. This impact is substantial: for 2-agent scenarios, increasing the sample size from $k = 3$ to $k = 20$ boosts the average welfare improvement by a remarkable 81.9% (from 2.1% to 3.82% over the baseline). This principle extends to the 3-agent setting, where increasing samples from $k = 10$ to $k = 20$ yields an additional 11.7% relative gain (from 10.37% to 11.58%). This finding underscores a key feature of our approach: the ability to explicitly control the trade-off between computational cost and economic efficiency.

### 4.4 INCENTIVE COMPATIBILITY: TRUTHFUL BIDDING IS APPROXIMATELY OPTIMAL

A central goal of our VCG-based mechanism is to incentivize truthful bidding, where each agent maximizes their utility by bidding their true value. This means an advertiser's optimal bid $b_i$ should match their true value $m_i$. To test this, we simulate a range of bidding scenarios. For every combination of true values $(m_1, m_2)$, we fix agent 2's bidding to be truthful and allow agent 1 to bid a range of values. We sweep Agent 1's true value from 0.0 to 1.0 in steps of 0.1 and Agent 2's true value from 0.0 to 1.0 in steps of 0.25, covering a wide range of value scenarios. For each combination of submitted bid and true values, we evaluate the average regret, the relative utility loss from truthful bidding compared to the optimal bid. In particular, we calculate it as the raw utility difference divided by its utility under the truthful bid. Additionally, we compute the absolute difference between agent 1's truthful bid value and the utility-maximizing bid.

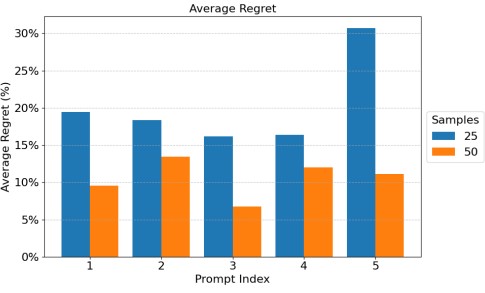
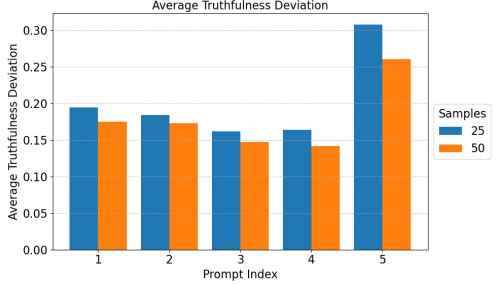

(a) Average regret down to 7% with $k = 50$ Monte Carlo samples.

(b) Average deviation down to 0.14 between true and optimal bid when $k = 50$.

Figure 7: Increasing Monte Carlo sample size improves truthfulness. Average regret and truthfulness deviation decrease from $k = 25$ to $k = 50$ samples, indicating truthfulness is approximately optimal.

Table 1: Image quality for 2 agents

| Agent 1 | Agent 2 | Mean | SD |
|---------|---------|--------|--------|
| 0.0 | 1.0 | 0.5081 | 0.0028 |
| 0.1 | 0.9 | 0.5084 | 0.0030 |
| 0.3 | 0.7 | 0.5083 | 0.0031 |
| 0.5 | 0.5 | 0.5082 | 0.0030 |
| 0.7 | 0.3 | 0.5085 | 0.0028 |
| 0.9 | 0.1 | 0.5090 | 0.0027 |
| 1.0 | 0.0 | 0.5089 | 0.0028 |

Table 2: Image quality for 3 agents

| Agent 1 | Agent 2 | Agent 3 | Mean | SD |
|---------|---------|---------|--------|--------|
| 0.33 | 0.33 | 0.33 | 0.5084 | 0.0029 |
| 0.40 | 0.40 | 0.20 | 0.5083 | 0.0030 |
| 0.60 | 0.30 | 0.10 | 0.5082 | 0.0029 |
| 0.60 | 0.20 | 0.20 | 0.5086 | 0.0029 |
| 1.00 | 0.00 | 0.00 | 0.5088 | 0.0027 |

**Results.** Figure 7a shows the average regret experienced by Agent 1 across five prompts. We present results with $k \in \{25, 50\}$ Monte Carlo samples and demonstrate that average regret consistently decreases with increased sampling. For $k = 50$, average regret remains below 15% across all prompts, reaching as low as 7% for individual prompts. This trend indicates that agents have reduced incentive to deviate from truthful bidding, further supporting the robustness of our mechanism under sampling-based generation.

Figure 7b reports the average deviation across all bid combinations between Agent 1's true bid value and the utility-maximizing bid across five prompts. Across all prompts, the average truthfulness deviation consistently decreases as $k$ increases. For $k = 50$, the average truthfulness deviation decreases to as low as 0.14. These findings provide empirical support that our mechanism preserves the truthfulness property of VCG in an approximate sense. Despite relying on Monte Carlo estimates of welfare during image generation, the learned auction dynamics can still incentivize agents to bid in alignment with their true values.

### 4.5 IMAGE QUALITY PRESERVATION

When composing multiple agent preferences, a fundamental concern is whether image quality degrades. To quantitatively evaluate this at scale, we use the CLIP alignment between generated images and the prompt "High quality photo" as a proxy for visual quality. We acknowledge that this is not a perfect measure of human aesthetic preference but use it here to verify that our mechanism does not systematically reduce quality. As shown in Tables 1 and 2, the average image quality remains stable at approximately 0.508 across all bid combinations for both 2-agent and 3-agent scenarios. This result indicates that our mechanism can blend multiple agent inputs without compromising visual fidelity.

## 5 DISCUSSION

**Applications** The framework we present offers strong potential for real-world applications, particularly in online advertising. Consider a search engine displaying banner ads on web pages: using our auction, it can generate visually compelling, contextually relevant images that reflect the preferences of multiple advertisers simultaneously. Our finding—that this generative auction yields higher welfare than single-winner auctions—provides a compelling incentive for platforms to adopt such methods. Advertisers benefit from having their branding or products meaningfully integrated into the generated content, with visibility and influence aligned with their bids. Beyond static images, our approach extends naturally to dynamic media. For example, a video platform could use our mechanism to embed branded elements, enabling multiple agents to bid for subtle, context-aware product placement. This opens new possibilities for non-intrusive, personalized advertising across multimedia formats.

**Limitations** A fundamental limitation of our approach arises from the inherent probabilistic nature of denoising diffusion models. This stochasticity, coupled with the potential for variability in how text prompts are interpreted by the diffusion model, means that we cannot analytically optimize the generated image to perfectly maximize welfare for a given set of bids. Our utilization of Monte Carlo sampling to estimate welfare and VCG payments is a direct consequence of this intractability. While increasing the number of samples leads to improved welfare and closer adherence to truthfulness,

this comes at the cost of increased computational resources and generation time, creating a trade-off between accuracy and efficiency.

**Future work.**   Our work provides the first empirical validation for generative image auctions with up to three agents. Exciting future directions include designing more sophisticated bidding languages that go beyond a single prompt and value, exploring revenue-maximizing versions of the auction, and adapting this mechanism to other generative modalities like video.

## 6   Conclusion

Our work presents the first generative image auction, using diffusion models to produce a single, multi-winner image that reflects aggregated agent preferences. We introduce an expressive score composition technique that enables nuanced, bid-dependent control over generated content, in conjunction with Monte Carlo VCG-based payments designed to ensure incentive compatibility. Our comprehensive empirical analysis, conducted on a diverse dataset with up to three agents, provides compelling evidence for our mechanism's effectiveness, demonstrating bid monotonicity, improved social welfare over single-winner baselines, and approximate incentive compatibility. As a foundational step, our research lays the groundwork for future investigation in controllable, incentive-aligned image generation, offering a new lens through which to design and evaluate generative systems.

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

# A  PROMPTS

Table 3: Prompts configurations across 20 2-Agent, 3-Agent Scenarios

| Base Prompt | Agent 1 Prompt | Agent 2 Prompt | Agent 3 Prompt |
|---|---|---|---|
| Two friends chatting over coffee at a cafe | Starbucks mug | Apple MacBook Air | New York Times newspaper |
| People enjoying a sunny day at the beach | Wilson volleyball | Red Bull cooler | Banana Boat sunscreen |
| A person relaxing on a living-room sofa | Lay's potato chips | M&M's candy | Netflix show on TV |
| A hiker resting on a mountain trail | North Face backpack | Gatorade bottle | Garmin watch |
| A professional working at a desk in a bright office | Dell monitor | Deer Park water bottle | Moleskine notebook |
| A woman with a tray of food at a cafeteria | McDonald's fries | Pepsi can | Heinz ketchup packet |
| A person walking down a city street | Louis Vuitton handbag | H&M storefront | Nike sneakers |
| Friends gathered for a movie night at home | Doritos chips | Coca-Cola cans | Amazon Fire TV remote |
| A car driving along a scenic coastal highway | Tesla car | ExxonMobil sign | Goodyear tires |
| Skiers waiting in a ski-lift line | Patagonia jacket | GoPro camera | Beats wireless headphones |
| A man waiting at a bus stop | Adidas backpack | Samsung Galaxy phone | Sony headphones |
| A student studying in a university library | MacBook Pro laptop | Red Bull energy drink | Coca-Cola can |
| A chef cooking in a restaurant kitchen | KitchenAid stand mixer | Coca-Cola glass bottle | Heinz ketchup bottle |
| A couple shopping at a farmers market | Whole Foods tote bag | iPhone camera | Patagonia fleece jacket |
| A teenager gaming in their bedroom | PlayStation 5 controller | Monster Energy drink | Razer gaming headset |
| A businesswoman in an airport lounge | Louis Vuitton luggage | Starbucks coffee cup | iPad Pro tablet |
| A mechanic working in an auto garage | Ford truck | DeWalt power drill | Mobil 1 motor oil |
| A barista working in a trendy coffee shop | Starbucks coffee machine | Spotify playlist screen | Levi's denim apron |
| A fitness trainer at a modern gym | Under Armour workout shirt | Beats wireless headphones | Hydroflask water bottle |
| A doctor in a hospital break room | MacBook Air laptop | Starbucks coffee cup | Samsung tablet |

Example images for the prompts are provided in the supplemental materials folder.

## B GENERALIZED SCORE COMPOSITION

The aggregation function can be extended to $n$ agents ($s^*_{1,n}(x)$) by iteratively defining score compositions with one less agent ($s^*_{1,n-1}(x)$). Assume, without loss of generality, that agents 1 to $n$ submit bids in decreasing value: $b_1 \geq b_2 \geq \ldots \geq b_n$. Denote $b_j^{(0)} = b_j$. For iteration $i$, denote $b_j^{(i)} = \frac{b_j^{(i-1)}}{\sum_{k=1}^{n-i+1} b_k^{(i-1)}}$ as the normalized bids for iteration $i$ and $w^{(i)} = \left(2\sum_{j=1}^{n-i} b_j^{(i)} - 1\right)$ as the normalized weights. Denote $c_{1,n} = \{c_1, c_2, \ldots, c_n\}$ as the combined prompt from all agents, and $s^*_{1,1}(x) = s(x|c, c_1)$ as the "base case" with the score conditional on dominant agent 1's prompt and the base prompt. We can recursively define $s^*_{1,n}(x)$, separating the least dominant agent $n$ and weighting the score function for the remaining $n - 1$ agents.

$$s^*_{1,n}(x) = (1 - w^{(1)})\, s(x|c, c_{1,n}) + w^{(1)}\, s^*_{1,n-1}(x) \tag{6}$$

$$s^*_{1,j}(x) = (1 - w^{(n-j+1)})\, s(x|c, c_{1,j}) + w^{(n-j+1)}\, s^*_{1,j-1}(x) \tag{7}$$

Then, we derive the closed form equation for $s^*_{1,n}(x)$.

$$s^*_{1,n}(x) = \prod_{i=1}^{n-1} w^{(i)}\, s(x|c_1) + \sum_{i=2}^{n} \left[ \left(\prod_{j=1}^{n-i} w^{(j)}\right) (1 - w^{(n-i+1)})\, s(x|c_{1,i}) \right] \tag{8}$$

## C LINEAR AGGREGATION

We had first considered the linear aggregation function.

$$s_t^{(1,2)}(x) = (1 - w^{(1)})s_t(x|c, c_1) + (w^{(1)})s_t(x|c, c_2) \tag{9}$$

However, we observed that it did not yield compositional images. Instead, the objects were faded on top of one another, leading to visually incoherent outcomes (Figure 8).

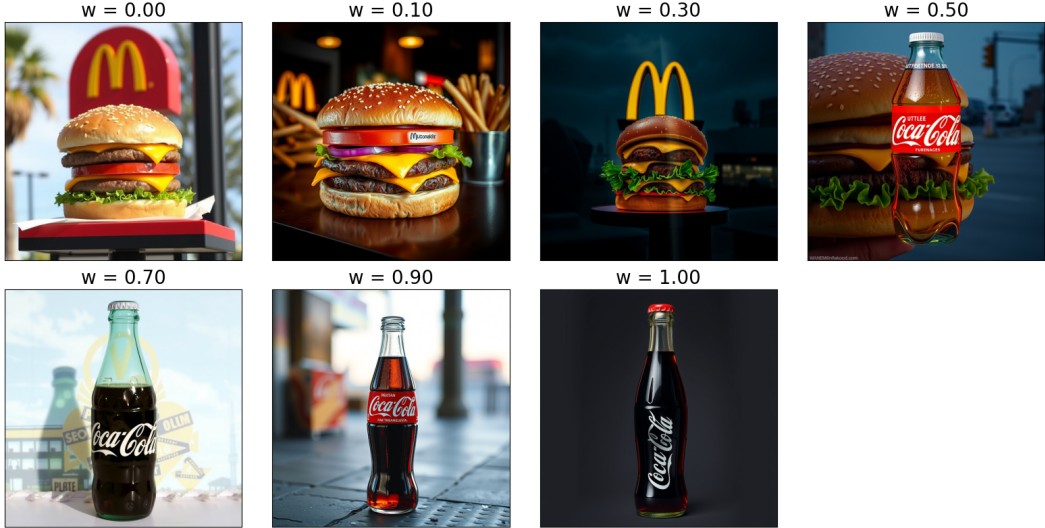

Figure 8: Linear aggregation function without the joint score fails to produce compositional images.

# D    COMPREHENSIVE EMPIRICAL EVALUATION

## D.1    LARGE-SCALE BIDDING

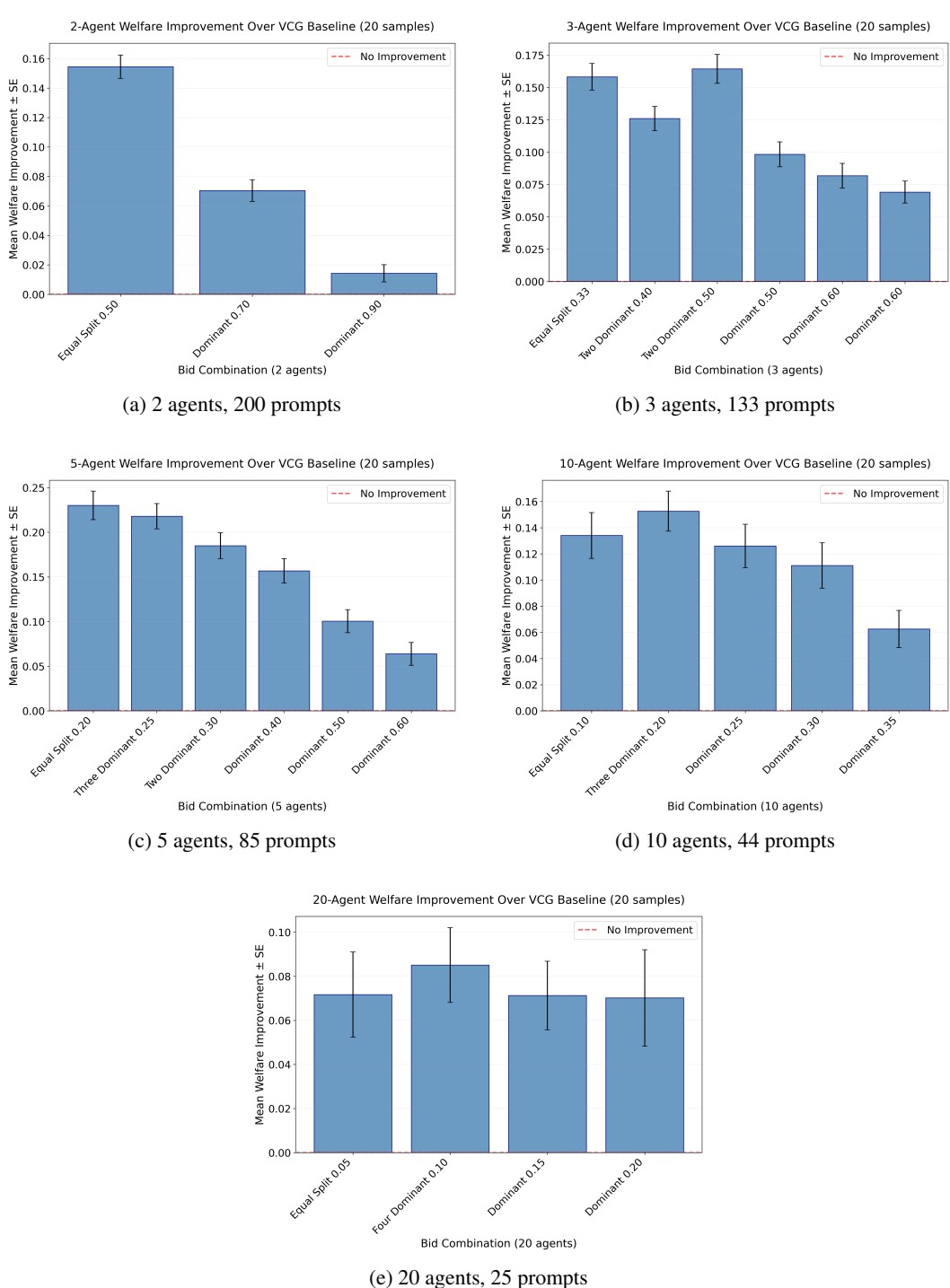

Figure 9: Welfare improvement over single-winner VCG baseline from 2 to 20 agents with varying bid combinations, achieving up to 23% increase in 5 agent setting.

## D.2 LAION IMAGE QUALITY

(a) 2 agents

(b) 3 agents

(c) 5 agents

(d) 10 agents

(e) 20 agents

Figure 10: LAION aesthetic quality scores for varying numbers of agents. Scores consistently remain above 5.75 across bidding combinations, indicating high image quality is preserved regardless of the number of agents or bid distribution.

### D.3 MCMC Samples Ablation

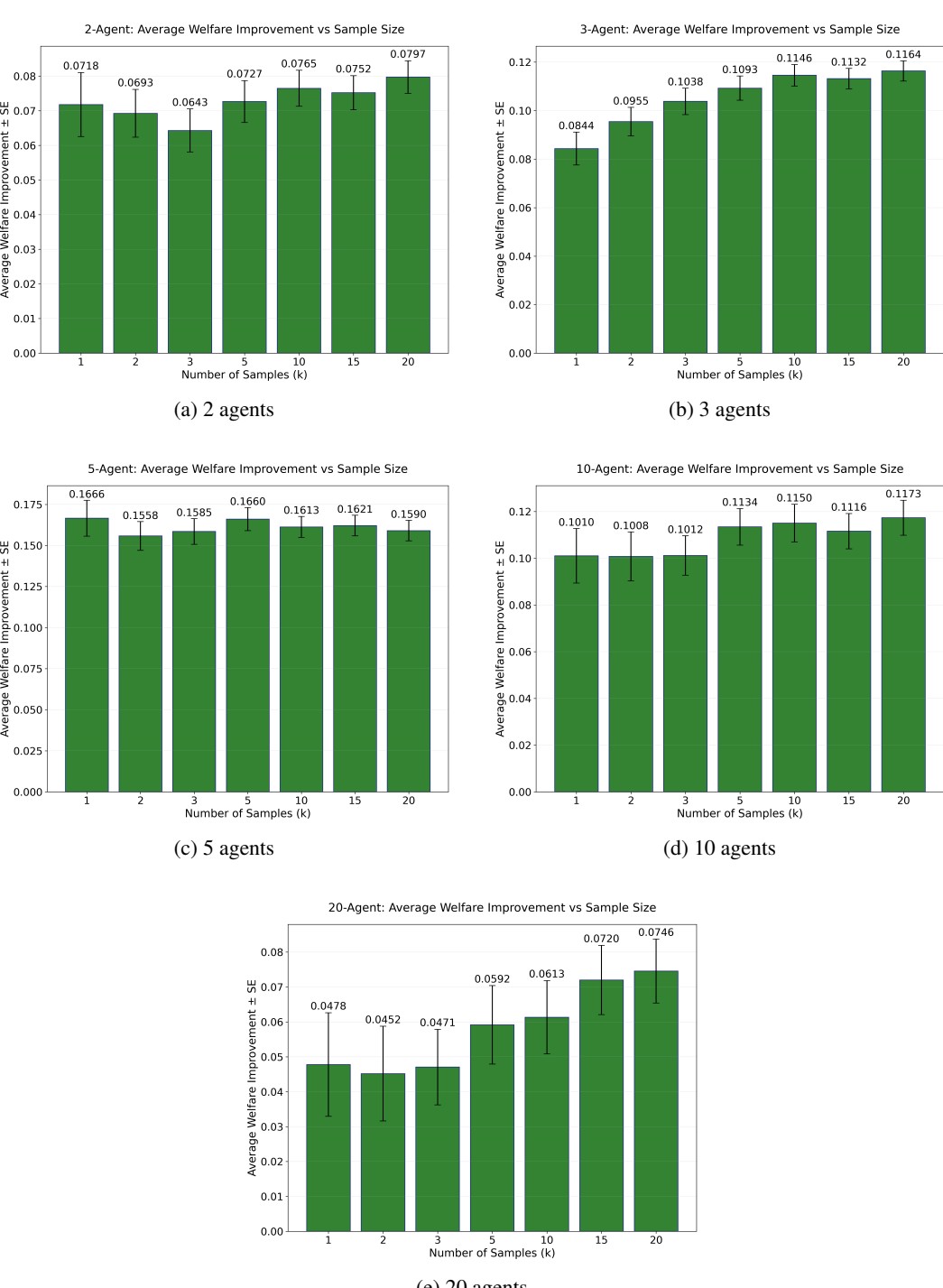

(a) 2 agents

(b) 3 agents

(c) 5 agents

(d) 10 agents

(e) 20 agents

Figure 11: Welfare improvement over single-winner VCG baseline as a function of Monte Carlo sample size (k). For 2, 3, 5, 10 agents, welfare improvement plateaus around $k = 5$ samples, reaching approximately 95% of near optimal welfare. For 20 agents, the plateau starts at $k = 15$ This demonstrates rapid convergence and practical scalability of our Monte Carlo approximation.

## D.4 COMPETITIVE BIDDING

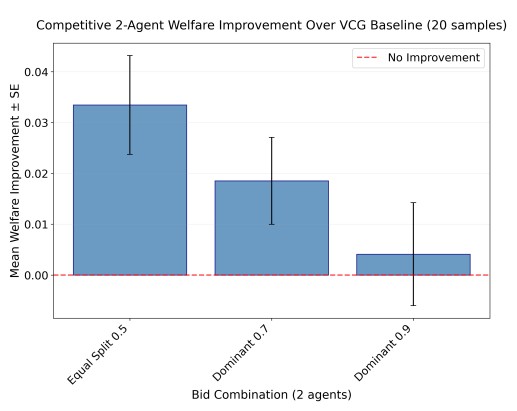

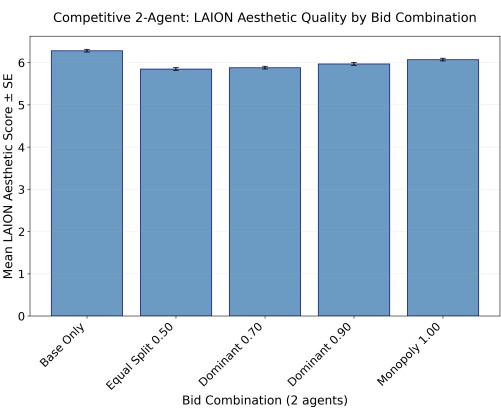

(a) Welfare improvement over single-winner baseline

(b) LAION image quality scores

Figure 12: Two agent competitive bidding results.

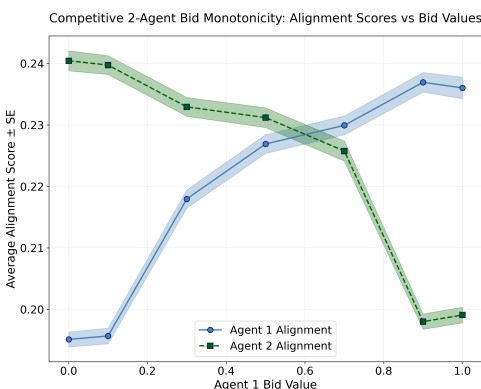

Figure 13: Bid monotonicity for two agent competitive bidding.

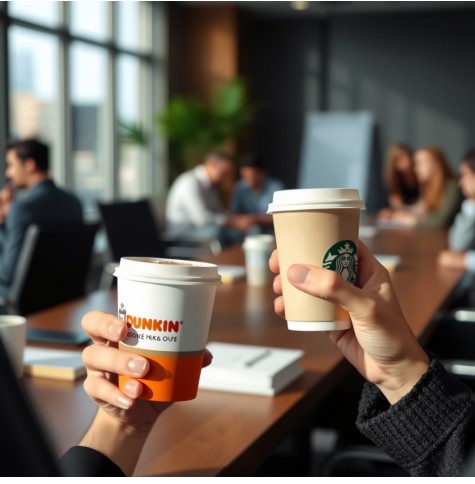

Figure 14: Example competitive bidding scenario with Dunkin and Starbucks (0.5/0.5).

864
865
866
867
868
869
870
871
872
873
874
875
876
877
878
879
880
881
882
883
884
885
886
887
888
889
890
891
892
893
894
895
896
897
898
899
900
901
902
903
904
905
906
907
908
909
910
911
912
913
914
915
916
917

Table 4: Prompts configurations across 20 2-Agent Competitive Scenarios

| Base Prompt | Agent 1 Prompt | Agent 2 Prompt |
| --- | --- | --- |
| A person drinking from a beverage can while working at a desk | Coca-Cola can | Pepsi can |
| Someone wearing athletic shoes while jogging in a park | Nike running shoes | Adidas running shoes |
| A person wearing a smartwatch while exercising at the gym | Apple Watch | Samsung Galaxy Watch |
| Someone drinking from a water bottle during a yoga class | Hydro Flask water bottle | Yeti water bottle |
| A driver filling up their car at a gas station | Shell gas station | Chevron gas station |
| A customer holding a shopping bag from a clothing store | Zara shopping bag | H&M shopping bag |
| A person wearing wireless earbuds while commuting | Apple AirPods | Samsung Galaxy Buds |
| Someone drinking an energy drink at a skateboard park | Red Bull can | Monster Energy can |
| A person carrying a backpack while hiking on a trail | North Face backpack | Patagonia backpack |
| Someone using a smartphone at a coffee shop | iPhone smartphone | Samsung Galaxy phone |
| A person wearing a baseball cap at a sports game | New York Yankees cap | Boston Red Sox cap |
| Someone drinking from a coffee cup in an office meeting | Starbucks coffee cup | Dunkin coffee cup |
| A customer carrying a fast food bag on a city street | McDonald's bag | Burger King bag |
| A person wearing running shorts during a marathon | Nike running shorts | Adidas running shorts |
| Someone eating chips while watching TV at home | Lay's chips bag | Doritos bag |
| A person wearing a winter jacket in a snowy setting | Canada Goose jacket | North Face jacket |
| Someone drinking orange juice at breakfast | Tropicana orange juice | Minute Maid orange juice |
| A person wearing a fitness tracker at a cycling class | Fitbit tracker | Garmin tracker |
| Someone holding a pizza box at a house party | Domino's pizza box | Pizza Hut box |
| A person wearing a hoodie at a college campus | Champion hoodie | Nike hoodie |

