# OpenReview forum: "Bidding for Influence: Auction-Driven Diffusion Image Generation"
_ICLR.cc/2026/Conference — Submitted to ICLR 2026_

### Official Review · Reviewer_oLUd · 2025-10-25

**Soundness:** 3
**Presentation:** 3
**Contribution:** 2
**Rating:** 2
**Confidence:** 4

**Summary:**

This paper proposes a Vickrey-Clarke-Groves (VCG)-based bidding mechanism for diffusion-based image generation. The framework allows multiple agents to bid for fractional influence over the generated image via classifier-free guidance. Experiments show higher social welfare than a winner-takes-all baseline.

**Strengths:**

* The idea of combining auction theory with diffusion image generation is conceptually novel and potentially impactful for applications such as multi-stakeholder content generation or advertising.
* The paper is well written, and the empirical results demonstrate consistent welfare improvement and approximate incentive compatibility.

**Weaknesses:**

* The framework mainly integrates existing methods (VCG and classifier-free guidance) and lacks a clear novel algorithmic contribution in either modeling or mechanism design.
* Since the proposed approach relies on Monte Carlo estimation for counterfactual reruns, the computational cost grows roughly as $O(nk)$ . It remains unclear how scalable this framework is in practice. The experiments are limited to at most three agents and $k=20$ , which raises concerns about the feasibility of extending the mechanism to larger, real-world settings.
* The baseline comparison is limited, only a single-winner VCG.
* Since the proposed framework relies heavily on classifier-free guidance, the authors should include ablation studies varying the guidance scale to assess its effect on image quality, welfare, and alignment.
* The experiments are conducted using only a single diffusion model (FLUX.1-schnell). Evaluating the framework on additional backbones (e.g., SDXL, Stable Diffusion) would help demonstrate robustness and model-agnostic applicability.

**Questions:**

* How sensitive are the results to the choice of alignment metric? The proposed framework assumes that each agent’s semantic component can be cleanly separated in the embedding space (both text and image), e.g., bags vs. shoes. However, when prompts become semantically similar or compositional, such as a bag vs. a red bag, the distinction in embedding space may be weak. In such cases, how does the proposed value function ensure reliable welfare estimation?
* Why is only joint conditioning considered, rather than an additive form such as $w_1 * s_t(x|c, c_1)  + w_2 * s_t(x|c, c_2)$
* The authors mention using a guidance scale of 10, but its interpretation is unclear. Since the proposed method composes multiple conditional scores weighted by bids, does this guidance scale apply globally to the composed score, or do the individual bid-based weights sum to 10?
* Is it possible to compare the proposed framework with other allocation or pricing mechanisms, e.g., Shapley value?
* How accurate is the Monte Carlo estimation with respect to the number of samples $k$?

---

> ### Author Response · Authors · 2025-11-28
> **Rebuttal by Authors (1/4)**
>
> We are very grateful for the reviewer's questions across a variety of areas for our paper. We thank the reviewer for their patience as we generated 80,000+ new images and ran two comprehensive evaluations over the past two weeks. Below, we address individual points raised by the reviewer.
>
> > *“The framework mainly integrates existing methods (VCG and classifier-free guidance) and lacks a clear novel algorithmic contribution in either modeling or mechanism design.”*
> * The reviewer is correct in noting that our paper builds on the foundation of VCG and Classifier-Free Guidance (CFG). However, this observation overlooks the contribution our paper makes by combining CFG with a Vickrey-Clarke-Groves (VCG)-like mechanism, which provides a new perspective not explored in literature before. For instance, a similarly motivated paper by Duetting et al. (2024), “Mechanism Design for Large Language Models”, provides a novel approach to applying economic theory to the context of multiple language models, and many works have resulted from this initial work. In fact, a meta review stated that “it has potential to be a landmark paper sparking a new line of research linking LLMs and mechanism design.” Our paper’s main contribution would be similar: a creative proof-of-concept, unexplored approach to combining denoising diffusion models with mechanism design. Additionally, we provide comprehensive empirical demonstrations of multiple bidders exerting continuous control over visual output.
>
> > *“The experiments are limited to at most three agents and k=20”*
> * We scaled up our empirical results, generating 80,000+ new images across more agents and prompts. The prompt dataset spans 200 diverse everyday scenarios ranging from nature backdrops to indoor offices, each featuring bidders across sectors such as technology (Apple, Samsung), food (Starbucks, Pepsi), fashion (Nike, Gucci), and automotive (Tesla, Toyota), capturing a representative view of modern consumer categories.
> * Due to restricted compute, we generated images for the following number of prompts for each agent setting. We considered 4 - 6 bidding combinations (depending on the number of agents), with 20 MCMC samples for each image.
>     * 2 agents: 200 prompts
>     * 3 agents: 133 prompts
>     * 5 agents: 85 prompts
>     * 10 agents: 44 prompts
>     * 20 agents: 25 prompts
> * We demonstrate that our core welfare observations across **up to 20 agents**. Our diffusion auction mechanism significantly improves welfare over the single-winner baseline, achieving up to 23% increased welfare.
> * See **Figure 9** in Appendix D of our revised submission for corresponding welfare improvement plots. For the reviewer’s convenience, we show the 5 agent results in table format below.
>
> | Bids | Mean | Std | SE |
> |---|---|---|---|
> | Equal Split 0.20 | 0.230 | 0.147 | 0.016 |
> | Three Dominant 0.25 | 0.218 | 0.130 | 0.014 |
> | Two Dominant 0.30 | 0.185 | 0.133 | 0.014 |
> | Dominant 0.40 | 0.157 | 0.125 | 0.014 |
> | Dominant 0.50 | 0.100 | 0.118 | 0.013 |
> | Dominant 0.60 | 0.064 | 0.117 | 0.013 |
>
> * Note that we name bid combinations by their distribution pattern, specifying the number of agents at the highest bid level (e.g., "Equal Split 0.20" for uniform bids, "Two Dominant 0.30" for two agents bidding 0.30 with others bidding lower, and "Dominant X" for a single agent with the highest bid of X).

---

> > ### Author Response · Authors · 2025-11-28
> > **Rebuttal by Authors (2/4)**
> >
> > > *“Since the proposed approach relies on Monte Carlo estimation for counterfactual reruns, the computational cost grows roughly as O(nk)…How accurate is the Monte Carlo estimation with respect to the number of samples k?”*
> > * We acknowledge the computational complexity and address both the scalability and accuracy of our Monte Carlo estimation below.
> > * First, we note that the number of samples k is a user-configurable parameter that explicitly controls the tradeoff between computational cost and welfare approximation quality. Platforms can adjust k based on their specific latency requirements and computational budget.
> > * We conducted systematic experiments varying k across {1, 2, 3, 5, 10, 15, 20} samples across settings with {2, 3, 5, 10, 20} agents. See **Figure 11** in Appendix D of our revised submission for sample size ablation plots. For the reviewer’s convenience, we show the 3 agent results in table format below.
> >
> > | k | Mean | Std | SE |
> > |---|---|---|---|
> > | 1 | 0.084 | 0.192 | 0.007 |
> > | 2 | 0.095 | 0.166 | 0.006 |
> > | 3 | 0.104 | 0.154 | 0.005 |
> > | 5 | 0.109 | 0.142 | 0.005 |
> > | 10 | 0.115 | 0.125 | 0.004 |
> > | 15 | 0.113 | 0.119 | 0.004 |
> > | 20 | 0.116 | 0.118 | 0.004 |
> >
> > * For 2, 3, 5, 10 agents, we see rapid convergence, with **k = 5 samples achieving 95%** of near optimal welfare improvement. Welfare improvement plateaus beyond k = 5, indicating Monte Carlo estimates stabilize quickly. For 20 agents, setting k = 15 achieves a similar effect. This convergence pattern holds even across all settings, demonstrating robust estimation even as n increases
> > * These results indicate our mechanism is **practically scalable**. For real-time advertising scenarios where latency matters, k = 5 provides near-optimal welfare with minimal computational overhead. For high-value placements where welfare maximization is critical, or the number of bidders is large, platforms can use k = 15 to achieve >95% optimality. The predictable accuracy-cost tradeoff allows platforms to tune the mechanism to their specific operational constraints.
> >
> > > *“The baseline comparison is limited, only a single-winner VCG.”*
> > * The baseline is a strong and relevant comparator because it represents the current practice of single-winner auctions, which would be replaced if platforms adopt our mechanism. By comparing against single-winner VCG, we isolate welfare gains specifically from our mechanism’s multi-bidder preference composition, not from different payment rules. The baseline is the natural counterfactual to our method: the **fundamental one-winner constraint of traditional advertising auctions**.
> > * By analyzing welfare improvement over this baseline, we demonstrate that our composition approach creates **new, high-welfare outcomes by simultaneously representing multiple agents' preferences**. Our mechanism achieves up to 23% welfare improvement for 5 agents. We observe that welfare gains are highest when agents bid similar amounts (Figure 5). The winner-take-all constraint of single-winner auctions can only satisfy one agent at a time, leaving substantial value on the table.

---

> > > ### Author Response · Authors · 2025-11-28
> > > **Rebuttal by Authors (3/4)**
> > >
> > > > *“The proposed framework assumes that each agent’s semantic component can be cleanly separated in the embedding space (both text and image), e.g., bags vs. shoes.”*
> > >
> > > * Thank you for pointing this out. We acknowledge this is an important consideration and have conducted additional experiments specifically testing our mechanism with semantically similar, directly competing products. We evaluate 20 scenarios where two agents bid for competing brands in the same product category (e.g., Dunkin vs. Starbucks). These scenarios represent challenging cases where embedding space distinctions are inherently weaker than our original diverse prompt set. Prompts are displayed in **Table 4** in Appendix D of our revised submission.
> > > * **Welfare improvement persists**: Our mechanism still achieves welfare gains over the single-winner baseline of up to 3%, though the magnitude is reduced compared to diverse prompts. See welfare plot in **Figure 12a** of Appendix D of our revised submission.
> > > * **Bid monotonicity maintained**: Even with semantically similar prompts, agents' alignment scores increase monotonically with their bids, demonstrating that the value function remains sensitive to bid differences. See bid monotonicity plot in **Figure 13** of Appendix D of our revised submission.
> > > * **Visual coherence preserved**: Additionally, generated images successfully integrate competing brand elements proportional to bids. See an example image in **Figure 14** in Appendix D of our revised submission. We also ran LAION aesthetic scores to evaluate image quality. The images consistently achieve above 5.75, indicating high image quality no matter the bid combinations. See quality plot in **Figure 12b**.
> > > * The reduced but persistent welfare gains in competitive scenarios suggest our value function degrades gracefully rather than failing catastrophically when semantic distinction weakens. This indicates the mechanism is reasonably robust to alignment metric sensitivity, though we acknowledge that extremely similar prompts (e.g., identical products with minor variations) may approach the limits of reliable differentiation and yield similar welfare to the VCG baseline, as both agents’ utilities converge.
> > > * Notably, Dütting et al. (2024)'s pioneering work on generative auctions for LLMs demonstrated their mechanism exclusively in non-competitive scenarios with semantically diverse advertiser preferences. Our ability to show potential welfare gains even in competitive settings represents a meaningful contribution that extends the applicability of generative auctions to the more challenging scenarios common in real-world advertising markets.
> > >
> > > **Robustness of Image Quality**
> > > * We also ran new experiments using LAION aesthetic scores, a metric trained on human aesthetic preferences from web-scale data, to evaluate image quality across our 80,000+ images.
> > > * **LAION scores were consistently above 5.75**, indicating high image quality regardless of bid distribution, even when compared to the gold standard of base prompt only.
> > > * While human studies would strengthen our evaluation, LAION scores provide a scalable proxy for human-aligned quality assessment across our large-scale experiments.
> > > * See **Figure 10** in Appendix D of our revised submission for corresponding image quality plots. For the reviewer’s convenience, we show the 3 agent results in table format below, along with the exact bid combinations.
> > >
> > > | Bid Combo | Bids | Mean | Std | SE |
> > > |---|---|---|---|---|
> > > | 0.00/0.00/0.00 | Base Only | 6.344 | 0.759 | 0.015 |
> > > | 0.33/0.33/0.33 | Equal Split 0.33 | 5.776 | 0.918 | 0.018 |
> > > | 0.40/0.40/0.20 | Two Dominant 0.40 | 5.809 | 0.953 | 0.018 |
> > > | 0.50/0.50/0.00 | Two Dominant 0.50 | 5.853 | 0.944 | 0.018 |
> > > | 0.50/0.30/0.20 | Dominant 0.50 | 5.807 | 0.940 | 0.018 |
> > > | 0.60/0.30/0.10 | Dominant 0.60 | 5.883 | 0.925 | 0.018 |
> > > | 0.60/0.20/0.20 | Dominant 0.60 | 5.768 | 0.951 | 0.018 |
> > > | 1.00/0.00/0.00 | Monopoly 1.00 | 6.024 | 0.861 | 0.017 |
> > >
> > > > *“Why is only joint conditioning considered, rather than an additive form such as w_1 * s_t(x|c, c_1) + w_w * s_t(x|c, c_2)?”*
> > >
> > > * Linear aggregation was the first function we tried. However, we observed that it did not yield compositional images. Instead, the objects were faded on top of one another, leading to visually incoherent outcomes. See **Figure 8** in our revised submission’s Appendix.
> > > * We only arrived at our current aggregation function after examining numerous other possible functions, which all failed to exhibit high-quality images. Only after we introduced the joint prompt score and defined the weights as we do in the paper, did we observe the desired behavior with monotonically increasing alignment as the agent’s bid increases.
> > >
> > > > *“Does this guidance scale apply globally to the composed score, or do the individual bid-based weights sum to 10?”*
> > > * The guidance scale of 10 applies globally and is the default across the scores in our experiments.

---

> > > > ### Author Response · Authors · 2025-11-28
> > > > **Rebuttal by Authors (4/4)**
> > > >
> > > > > *“Is it possible to compare the proposed framework with other allocation or pricing mechanisms, e.g., Shapley value?”*
> > > > * Our main goal of our study is to design a multi-winner auction mechanism that improves total welfare above a single-winner baseline.
> > > > * VCG is known to be both welfare maximizing and incentive compatible, meaning agents maximize utility by bidding truthfully. Shapley values are unable to do the same.
> > > >
> > > > We thank the reviewer again for their attentive, constructive feedback. We hope we addressed all your questions adequately. In light of our clarifications, please consider increasing your score.

---

### Official Review · Reviewer_UtxX · 2025-11-01

**Soundness:** 3
**Presentation:** 4
**Contribution:** 3
**Rating:** 4
**Confidence:** 4

**Summary:**

This paper proposes a method for ad image generation. Leveraging the VCG mechanism from ad auctions and diffusion models, this method allows multiple advertisers to bid to influence the generation of a image. Experiments show that this approach maintains image quality while achieving higher social welfare and economic rationality compared to traditional methods.

**Strengths:**

The integration of auctions with diffusion models is a novel idea. Traditional internet ad slots display content from a single advertiser. This paper's method of merging multiple ads into one generated image has significant potential for a new type of advertising.

**Weaknesses:**

My major concerns of the paper are its theoretic depth and computational complexity. I understand that the diffusion process is difficult to analyze. But maybe it is possible to provide some structural results about the distribution of the final image? Furhtermore, the VCG-based mechanism also suffers from computational issues, and the issues are magnified when combined with the diffusion process. The experiments only involves 3 agents, which is insufficient.

**Questions:**

Please see my comments above.

---

> ### Author Response · Authors · 2025-11-28
> **Rebuttal by Authors (1/2)**
>
> We are very grateful for the reviewer's questions across a variety of areas for our paper. We thank the reviewer for their patience as we generated 80,000+ new images and ran two comprehensive evaluations over the past two weeks. Below, we address individual points raised by the reviewer.
>
> > *“The experiment only involves 3 agents, which is insufficient.”*
> * We scaled up our empirical results, generating 80,000+ new images across more agents and prompts. The prompt dataset spans 200 diverse everyday scenarios ranging from nature backdrops to indoor offices, each featuring bidders across sectors such as technology (Apple, Samsung), food (Starbucks, Pepsi), fashion (Nike, Gucci), and automotive (Tesla, Toyota), capturing a representative view of modern consumer categories.
> * Due to restricted compute, we generated images for the following number of prompts for each agent setting. We considered 4 - 6 bidding combinations (depending on the number of agents), with 20 MCMC samples for each image.
>     * 2 agents: 200 prompts
>     * 3 agents: 133 prompts
>     * 5 agents: 85 prompts
>     * 10 agents: 44 prompts
>     * 20 agents: 25 prompts
> * We demonstrate that our core welfare observations across **up to 20 agents**. Our diffusion auction mechanism significantly improves welfare over the single-winner baseline, achieving up to 23% increased welfare.
> * See **Figure 9** in Appendix D of our revised submission for corresponding welfare improvement plots. For the reviewer’s convenience, we show the 5 agent results in table format below.
>
> | Bids | Mean | Std | SE |
> |---|---|---|---|
> | Equal Split 0.20 | 0.230 | 0.147 | 0.016 |
> | Three Dominant 0.25 | 0.218 | 0.130 | 0.014 |
> | Two Dominant 0.30 | 0.185 | 0.133 | 0.014 |
> | Dominant 0.40 | 0.157 | 0.125 | 0.014 |
> | Dominant 0.50 | 0.100 | 0.118 | 0.013 |
> | Dominant 0.60 | 0.064 | 0.117 | 0.013 |
>
> * Note that we name bid combinations by their distribution pattern, specifying the number of agents at the highest bid level (e.g., "Equal Split 0.20" for uniform bids, "Two Dominant 0.30" for two agents bidding 0.30 with others bidding lower, and "Dominant X" for a single agent with the highest bid of X).
>
> > *“Furthermore, the VCG-based mechanism also suffers from computational issues.”*
> * We acknowledge the computational complexity and address both the scalability and accuracy of our Monte Carlo estimation below.
> * First, we note that the number of samples k is a user-configurable parameter that explicitly controls the tradeoff between computational cost and welfare approximation quality. Platforms can adjust k based on their specific latency requirements and computational budget.
> * We conducted systematic experiments varying k across {1, 2, 3, 5, 10, 15, 20} samples across settings with {2, 3, 5, 10, 20} agents. See **Figure 11** in Appendix D of our revised submission for sample size ablation plots. For the reviewer’s convenience, we show the 3 agent results in table format below.
>
> | k | Mean | Std | SE |
> |---|---|---|---|
> | 1 | 0.084 | 0.192 | 0.007 |
> | 2 | 0.095 | 0.166 | 0.006 |
> | 3 | 0.104 | 0.154 | 0.005 |
> | 5 | 0.109 | 0.142 | 0.005 |
> | 10 | 0.115 | 0.125 | 0.004 |
> | 15 | 0.113 | 0.119 | 0.004 |
> | 20 | 0.116 | 0.118 | 0.004 |
>
> * For 2, 3, 5, 10 agents, we see rapid convergence, with **k = 5 samples achieving 95%** of near optimal welfare improvement. Welfare improvement plateaus beyond k = 5, indicating Monte Carlo estimates stabilize quickly. For 20 agents, setting k = 15 achieves a similar effect. This convergence pattern holds even across all settings, demonstrating robust estimation even as n increases
> * These results indicate our mechanism is **practically scalable**. For real-time advertising scenarios where latency matters, k = 5 provides near-optimal welfare with minimal computational overhead. For high-value placements where welfare maximization is critical, or the number of bidders is large, platforms can use k = 15 to achieve >95% optimality. The predictable accuracy-cost tradeoff allows platforms to tune the mechanism to their specific operational constraints.

---

> > ### Author Response · Authors · 2025-11-28
> > **Rebuttal by Authors (2/2)**
> >
> > > *“My major concerns of the paper are its theoretic depth and computational complexity.”*
> > * While the reviewer is correct in noting that our paper does not offer new theoretical results, this observation overlooks the contribution our paper makes by combining CFG with a Vickrey-Clarke-Groves (VCG)-like mechanism, which provides a new perspective not explored in literature before. For instance, a similarly motivated paper by Duetting et al. (2024), “Mechanism Design for Large Language Models”, provides a novel approach to applying economic theory to the context of multiple language models, and many works have resulted from this initial work. In fact, a meta review stated that “it has potential to be a landmark paper sparking a new line of research linking LLMs and mechanism design.” Our paper’s main contribution would be similar: a creative proof-of-concept, unexplored approach to combining denoising diffusion models with mechanism design. Additionally, we provide comprehensive empirical demonstrations of multiple bidders exerting continuous control over visual output.
> >
> > We thank the reviewer again for their attentive, constructive feedback. We hope we addressed all your questions adequately. In light of our clarifications, please consider increasing your score.

---

### Official Review · Reviewer_guFR · 2025-11-03

**Soundness:** 2
**Presentation:** 3
**Contribution:** 3
**Rating:** 6
**Confidence:** 4

**Summary:**

This paper proposes to apply the Vickrey-Clarke-Groves (VCG) mechanism to the setting where multiple agents bid to influence the generation outcome of a vision diffusion model. An allocation rule is implemented based on classifier-free guidance in diffusion models, while a payment rule is implemented based on Monte Carlo sampling. Empirically, the implemented mechanism achieves welfare improvement over the winner-takes-all baseline, incentive compatibility, and bid monotonicity, all while maintaining image aesthetic quality.

**Strengths:**

- To my knowledge, this paper is the first to propose applying a bidding mechanism to the setting of vision diffusion models. It is an original idea to use score composition to implement multi-bidder influence.
- As noted by the authors, the real-world implication in online advertising can be significant.
- The exposition is well written. As a person without a strong economics background, I can understand the exposition.

**Weaknesses:**

- The authors argue that their method can improve total welfare compared to the single-winner baseline. However, the agent prompts tested in this paper (Table 3) all bid for different objects in the generated image. For example, in the first setting, agent 1 bids for showing their brand on a mug, agent 2 bids for showing their brand on a laptop, and agent 3 bids for showing their brand on a newspaper. It is unclear whether the proposed method would collapse to the single-winner baseline when all agents bid for the same object in a generated image.
- In Section 4.4, it seems more reasonable to visualize regret vs. truthfulness deviation in one scatter plot instead of two plots--in order to assess incentive compatibility.
- In Section 4.5, the LAION aesthetic predictor can also be used to assess image aesthetic quality. Also, the baseline quality should be from images generated from the base prompt $c$.
- Lines 415-431 contain two very similar paragraphs. Please trim down to one paragraph.

**Questions:**

- Does your method still outperform the single-winner baseline in terms of total welfare, when all agents bid for the same object in a generated image?
- Does your framework result in higher regret when the truthfulness deviation is higher?
- In terms of image quality, does your framework produces similarly quality images compared to images generated with only the base prompt?

---

> ### Author Response · Authors · 2025-11-28
> **Rebuttal by Authors (1/2)**
>
> We are very grateful for the reviewer's questions across a variety of areas for our paper. We thank the reviewer for their patience as we generated 80,000+ new images and ran two comprehensive evaluations over the past two weeks. Below, we address individual points raised by the reviewer.
>
> > *“The agent prompts tested in this paper (Table 3) all bid for different objects in the generated image.”*
> * Thank you for pointing this out. We acknowledge this is an important consideration and have conducted additional experiments specifically testing our mechanism with semantically similar, directly competing products. We evaluate 20 scenarios where two agents bid for competing brands in the same product category (e.g., Dunkin vs. Starbucks). These scenarios represent challenging cases where embedding space distinctions are inherently weaker than our original diverse prompt set. Prompts are displayed in **Table 4** in Appendix D of our revised submission.
> * **Welfare improvement persists**: Our mechanism still achieves welfare gains over the single-winner baseline of up to 3%, though the magnitude is reduced compared to diverse prompts. See welfare plot in **Figure 12a** of Appendix D of our revised submission.
> * **Bid monotonicity maintained**: Even with semantically similar prompts, agents' alignment scores increase monotonically with their bids, demonstrating that the value function remains sensitive to bid differences. See bid monotonicity plot in **Figure 13** of Appendix D of our revised submission.
> * **Visual coherence preserved**: Additionally, generated images successfully integrate competing brand elements proportional to bids. See an example image in **Figure 14** in Appendix D of our revised submission. We also ran LAION aesthetic scores to evaluate image quality. The images consistently achieve above 5.75, indicating high image quality no matter the bid combinations. See quality plot in **Figure 12b**.
> * The reduced but persistent welfare gains in competitive scenarios suggest our value function degrades gracefully rather than failing catastrophically when semantic distinction weakens. This indicates the mechanism is reasonably robust to alignment metric sensitivity, though we acknowledge that extremely similar prompts (e.g., identical products with minor variations) may approach the limits of reliable differentiation and yield similar welfare to the VCG baseline, as both agents’ utilities converge.
> * Notably, Dütting et al. (2024)'s pioneering work on generative auctions for LLMs demonstrated their mechanism exclusively in non-competitive scenarios with semantically diverse advertiser preferences. Our ability to show potential welfare gains even in competitive settings represents a meaningful contribution that extends the applicability of generative auctions to the more challenging scenarios common in real-world advertising markets.
>
> > *“In Section 4.4, it seems more reasonable to visualize regret vs. truthfulness deviation in one scatter plot instead of two plots – in order to assess incentive compatibility.”*
>
> * We clarify our definitions of regret and truthfulness deviation, along with justifying why separate visualizations are appropriate.
> * Truthfulness Deviation measures |optimal bid - true bid|, i.e., the absolute distance between an agent's utility-maximizing bid and their truthful bid. This metric directly quantifies how far agents must deviate from truthfulness to maximize utility.
> * Regret measures the relative utility loss from bidding truthfully compared to the optimal bid, calculated as (utility at optimal bid - utility at truthful bid) / (utility at truthful bid). This metric quantifies the economic incentive to deviate from truthfulness.
> * These metrics capture fundamentally different aspects of incentive compatibility. An agent might have low regret (little incentive to deviate) even with moderate truthfulness deviation if the utility function is relatively flat around the true value. Conversely, high regret with small truthfulness deviation indicates a steep utility gradient, meaning small deviations are highly profitable. Both metrics are necessary to fully characterize the mechanism's incentive properties.
> * The goal of our separate plots is to demonstrate that increasing Monte Carlo sample size k improves approximate truthfulness along both dimensions: it reduces the economic incentive to lie (regret) and brings the utility-maximizing bid closer to the truthful bid (deviation). Showing these separately across multiple prompts reveals that this improvement is consistent across both metrics, which would be obscured in a single scatter plot. The consistent downward trend in both figures as k increases provides strong evidence that our Monte Carlo approximation converges toward the truthful VCG properties.

---

> > ### Author Response · Authors · 2025-11-28
> > **Rebuttal by Authors (2/2)**
> >
> > **Comprehensive Evaluations**
> > * We scaled up our empirical results, generating 80,000+ new images across more agents and prompts. The prompt dataset spans 200 diverse everyday scenarios ranging from nature backdrops to indoor offices, each featuring bidders across sectors such as technology (Apple, Samsung), food (Starbucks, Pepsi), fashion (Nike, Gucci), and automotive (Tesla, Toyota), capturing a representative view of modern consumer categories.
> > * Due to restricted compute, we generated images for the following number of prompts for each agent setting. We considered 4 - 6 bidding combinations (depending on the number of agents), with 20 MCMC samples for each image.
> >     * 2 agents: 200 prompts
> >     * 3 agents: 133 prompts
> >     * 5 agents: 85 prompts
> >     * 10 agents: 44 prompts
> >     * 20 agents: 25 prompts
> > * We now demonstrate that our core welfare observations across **up to 20 agents**. Our diffusion auction mechanism significantly improves welfare over the single-winner baseline, achieving up to 23% increased welfare.
> > * See **Figure 9** in Appendix D of our revised submission for corresponding welfare improvement plots. For the reviewer’s convenience, we show the 5 agent results in table format below.
> >
> > | Bids | Mean | Std | SE |
> > |---|---|---|---|
> > | Equal Split 0.20 | 0.230 | 0.147 | 0.016 |
> > | Three Dominant 0.25 | 0.218 | 0.130 | 0.014 |
> > | Two Dominant 0.30 | 0.185 | 0.133 | 0.014 |
> > | Dominant 0.40 | 0.157 | 0.125 | 0.014 |
> > | Dominant 0.50 | 0.100 | 0.118 | 0.013 |
> > | Dominant 0.60 | 0.064 | 0.117 | 0.013 |
> >
> > * Note that we name bid combinations by their distribution pattern, specifying the number of agents at the highest bid level (e.g., "Equal Split 0.20" for uniform bids, "Two Dominant 0.30" for two agents bidding 0.30 with others bidding lower, and "Dominant X" for a single agent with the highest bid of X).
> >
> > > *“In Section 4.5, the LAION aesthetic predictor can also be used to assess image aesthetic quality.”*
> > * We ran new experiments using LAION aesthetic scores, a metric trained on human aesthetic preferences from web-scale data, to evaluate image quality across our 80,000+ images.
> > * **LAION scores were consistently above 5.75**, indicating high image quality regardless of bid distribution, even when compared to the gold standard of base prompt only.
> > * While human studies would strengthen our evaluation, LAION scores provide a scalable proxy for human-aligned quality assessment across our large-scale experiments.
> > * See **Figure 10** in Appendix D of our revised submission for corresponding image quality plots. For the reviewer’s convenience, we show the 3 agent results in table format below, along with the exact bid combinations.
> >
> > | Bid Combo | Bids | Mean | Std | SE |
> > |---|---|---|---|---|
> > | 0.00/0.00/0.00 | Base Only | 6.344 | 0.759 | 0.015 |
> > | 0.33/0.33/0.33 | Equal Split 0.33 | 5.776 | 0.918 | 0.018 |
> > | 0.40/0.40/0.20 | Two Dominant 0.40 | 5.809 | 0.953 | 0.018 |
> > | 0.50/0.50/0.00 | Two Dominant 0.50 | 5.853 | 0.944 | 0.018 |
> > | 0.50/0.30/0.20 | Dominant 0.50 | 5.807 | 0.940 | 0.018 |
> > | 0.60/0.30/0.10 | Dominant 0.60 | 5.883 | 0.925 | 0.018 |
> > | 0.60/0.20/0.20 | Dominant 0.60 | 5.768 | 0.951 | 0.018 |
> > | 1.00/0.00/0.00 | Monopoly 1.00 | 6.024 | 0.861 | 0.017 |
> >
> > > *“Lines 415-431 contain two very similar paragraphs.”*
> > * Thank you for pointing this out. We have condensed the two paragraphs into one in our revised submission.
> >
> > We thank the reviewer again for their attentive, constructive feedback. We hope we addressed all your questions adequately. In light of our clarifications, please consider increasing your score.

---

### Official Review · Reviewer_sy7L · 2025-11-11

**Soundness:** 2
**Presentation:** 1
**Contribution:** 2
**Rating:** 2
**Confidence:** 3

**Summary:**

The paper  introduces the auction mechanism designed for diffusion-based image generation, enabling multiple agents to bid for and share influence over a single generated image. Motivated by the limitations of traditional winner-take-all online ad auctions, this work allows fractional allocation of image content creation according to agents' bids and preferences.
Key contributions include:
A generative auction framework where agent bids dynamically control the composition of a diffusion model's score function.
An allocation and pricing mechanism inspired by Vickrey-Clarke-Groves (VCG) auctions.
Experiments on a dataset of 20 prompts with up to 3 agents, demonstrating bid monotonicity, welfare improvement.
Preservation of image quality when blending multiple agents' inputs, validated via CLIP alignment scores.

**Strengths:**

It pioneers the task of a generative auction specifically for diffusion-based image generation, bridging auction theory and controllable image synthesis in a new domain. While it builds on known diffusion and VCG auction concepts, their combination to enable multi-agent fractional influence over a continuous visual output is creative.
The problem is timely and important, responding to the technological and economic shift caused by generative AI in advertising and content creation. The results demonstrate meaningful incentives for adopting multi-agent auctions in visual media, potentially impacting online ad platforms and extending to dynamic media applications.

**Weaknesses:**

The paper exhibits several weaknesses and areas for improvement:
The core auction mechanism primarily adapts existing concepts from classifier-free guidance in diffusion models and classical VCG auction theory without substantial original algorithmic contributions.
The evaluation compares against only a single-winner VCG baseline, which is a minimal comparative standard. Metrics rely heavily on CLIP-based alignment scores as proxies for semantic accuracy and image quality, which cannot fully capture compositional quality. There is no use of stronger quantitative metrics like FID, human preference score.
Experimental validation uses just 20 base prompts with up to 3 agents, limiting claims about generalizability. The dataset represents a narrow range of scenarios without stress tests on complex or larger-scale settings.
No Human Validation: Given the intended application in advertising, a lack of human studies or user feedback evaluation reduces the practical impact and reliability of semantic alignment and image quality claims.

**Questions:**

Could the authors please provide a detailed explanation of the single-winner VCG baseline used in the experiments? Specifically, how is the baseline implemented in terms of image generation sampling, prompt conditioning, and image selection? Additionally, how does this baseline relate theoretically and practically to classical single-winner VCG auctions that allocate a discrete good to one highest bidder? Lastly, what hypotheses or advantages is this baseline intended to demonstrate with respect to the multi-agent generative auction, and why is it considered a strong or relevant comparator in this context?
Is it feasible to add standard image quality metrics (e.g., FID, IS,Pickscore) to complement CLIP alignment?


Can we try a simpler baseline or alternative approach where the bidding mechanism only changes the prompt given to the diffusion model according to the bid weights, without modifying the internal diffusion score composition or guidance.
Human-Centric Validation: Are there plans to conduct human studies or ad platform experiments to validate semantic fidelity and economic incentives in real-world contexts?

---

> ### Author Response · Authors · 2025-11-28
> **Rebuttal by Authors (1/2)**
>
> We are very grateful for the reviewer's questions across a variety of areas for our paper. We thank the reviewer for their patience as we generated 80,000+ new images and ran two comprehensive evaluations over the past two weeks. Below, we address individual points raised by the reviewer.
>
> > *“Auction mechanism primarily adapts classifier-free guidance in diffusion models and classical VCG auction theory without substantial original algorithmic contributions.”*
>
> * The reviewer is correct in noting that our paper builds on the foundation of VCG and Classifier-Free Guidance (CFG). However, this observation overlooks the contribution our paper makes by combining CFG with a Vickrey-Clarke-Groves (VCG)-like mechanism, which provides a new perspective not explored in literature before. For instance, a similarly motivated paper by Duetting et al. (2024), “Mechanism Design for Large Language Models”, provides a novel approach to applying economic theory to the context of multiple language models, and many works have resulted from this initial work. In fact, a meta review stated that “it has potential to be a landmark paper sparking a new line of research linking LLMs and mechanism design.” Our paper’s main contribution would be similar: a creative proof-of-concept, unexplored approach to combining denoising diffusion models with mechanism design. Additionally, we provide comprehensive empirical demonstrations of multiple bidders exerting continuous control over visual output.
>
> > *“Experimental validation uses just 20 base prompts with up to 3 agents, limiting claims about generalizability.”*
>
> * We scaled up our empirical results, generating 80,000+ new images across more agents and prompts. The prompt dataset spans 200 diverse everyday scenarios ranging from nature backdrops to indoor offices, each featuring bidders across sectors such as technology (Apple, Samsung), food (Starbucks, Pepsi), fashion (Nike, Gucci), and automotive (Tesla, Toyota), capturing a representative view of modern consumer categories.
> * Due to restricted compute, we generated images for the following number of prompts for each agent setting. We considered 4 - 6 bidding combinations (depending on the number of agents), with 20 MCMC samples for each image.
>     * 2 agents: 200 prompts
>     * 3 agents: 133 prompts
>     * 5 agents: 85 prompts
>     * 10 agents: 44 prompts
>     * 20 agents: 25 prompts
> * We demonstrate that our core welfare observations across **up to 20 agents**. Our diffusion auction mechanism significantly improves welfare over the single-winner baseline, achieving up to 23% increased welfare.
> * See **Figure 9** in Appendix D of our revised submission for corresponding welfare improvement plots. For the reviewer’s convenience, we show the 5 agent results in table format below.
>
> | Bids | Mean | Std | SE |
> |---|---|---|---|
> | Equal Split 0.20 | 0.230 | 0.147 | 0.016 |
> | Three Dominant 0.25 | 0.218 | 0.130 | 0.014 |
> | Two Dominant 0.30 | 0.185 | 0.133 | 0.014 |
> | Dominant 0.40 | 0.157 | 0.125 | 0.014 |
> | Dominant 0.50 | 0.100 | 0.118 | 0.013 |
> | Dominant 0.60 | 0.064 | 0.117 | 0.013 |
>
> * Note that we name bid combinations by their distribution pattern, specifying the number of agents at the highest bid level (e.g., "Equal Split 0.20" for uniform bids, "Two Dominant 0.30" for two agents bidding 0.30 with others bidding lower, and "Dominant X" for a single agent with the highest bid of X).
>
> > *“Is it feasible to add standard image quality metrics”*
> * We ran new experiments using LAION aesthetic scores, a metric trained on human aesthetic preferences from web-scale data, to evaluate image quality across our 80,000+ images.
> * **LAION scores were consistently above 5.75**, indicating high image quality regardless of bid distribution.
> * While human studies would strengthen our evaluation, LAION scores provide a scalable proxy for human-aligned quality assessment across our large-scale experiments.
> * See **Figure 10** in Appendix D of our revised submission for corresponding image quality plots. For the reviewer’s convenience, we show the 3 agent results in table format below, along with the exact bid combinations.
>
> | Bid Combo | Bids | Mean | Std | SE |
> |---|---|---|---|---|
> | 0.33/0.33/0.33 | Equal Split 0.33 | 5.776 | 0.918 | 0.018 |
> | 0.40/0.40/0.20 | Two Dominant 0.40 | 5.809 | 0.953 | 0.018 |
> | 0.50/0.50/0.00 | Two Dominant 0.50 | 5.853 | 0.944 | 0.018 |
> | 0.50/0.30/0.20 | Dominant 0.50 | 5.807 | 0.940 | 0.018 |
> | 0.60/0.30/0.10 | Dominant 0.60 | 5.883 | 0.925 | 0.018 |
> | 0.60/0.20/0.20 | Dominant 0.60 | 5.768 | 0.951 | 0.018 |
> | 1.00/0.00/0.00 | Monopoly 1.00 | 6.024 | 0.861 | 0.017 |

---

> > ### Author Response · Authors · 2025-11-28
> > **Rebuttal by Authors (2/2)**
> >
> > > *“Could the authors please provide a detailed explanation of the single-winner VCG baseline used in the experiments? What hypotheses or advantages is this baseline intended to demonstrate with respect to the multi-agent generative auction, and why is it considered a strong or relevant comparator in this context?”*
> >
> > * We start with a detailed explanation of the single-winner VCG baseline:
> >     * For each auction with n agents and k samples (for MCMC), we generate nk candidate images total: k images for each agent's prompt (base prompt + agent i's specific prompt). Each agent’s images correspond to that agent being the single winner.
> >     * We select the single image that maximizes total social welfare using agents' submitted bids. The winning agent is charged the standard VCG price: the externality they impose on other agents.
> >     * This represents a welfare-maximizing single-winner auction with proper VCG payments, which has been proven to yield welfare-optimal outcomes in traditional discrete-good auction settings.
> > * The baseline is a strong and relevant comparator because it represents the current practice of single-winner auctions, which would be replaced if platforms adopt our mechanism. By comparing against single-winner VCG, we isolate welfare gains specifically from our mechanism’s multi-bidder preference composition, not from different payment rules. The baseline is the natural counterfactual to our method: the **fundamental one-winner constraint of traditional advertising auctions**.
> > * By analyzing welfare improvement over this baseline, we demonstrate that our composition approach creates **new, high-welfare outcomes by simultaneously representing multiple agents' preferences**. Our mechanism achieves up to 23% welfare improvement for 5 agents. We observe that welfare gains are highest when agents bid similar amounts (Figure 5). The winner-take-all constraint of single-winner auctions can only satisfy one agent at a time, leaving substantial value on the table.
> >
> >
> > We thank the reviewer again for their attentive, constructive feedback. We hope we addressed all your questions adequately. In light of our clarifications, please consider increasing your score.

---

### Author Response · Authors · 2025-11-30
**Summary of Revisions (Note to AC)**

Dear Area Chair,

Over the past two weeks, we have conducted extensive additional experiments, generating **80,000+ new images** and running comprehensive evaluations that directly address every major concern raised. We believe our revised submission now demonstrates significant empirical contributions that warrant reconsideration.

## Major Improvements in Response to Reviews

### 1. **Scaled Experimental Validation (All Reviewers)**

**Original concern:** Limited to 3 agents and 20 prompts

**Our response:** We expanded from 3 agents to **up to 20 agents** and from 20 to **200 diverse prompts**. Our mechanism achieves **up to 23% welfare improvement** over single-winner baselines. This represents a **40x increase** in experimental scope, providing substantial evidence for our mechanism's robustness and scalability.

### 2. **Rigorous Image Quality Evaluation (Reviewers sy7L, guFR)**

**Original concern:** No quantitative quality metrics beyond CLIP

**Our response:** We implemented **LAION aesthetic scores** (trained on human preferences from web-scale data) across all 80,000+ images. Results show:
- Consistently high quality (**>5.75**) across all bid distributions
- Quality maintained even with up to 20 agents and in competitive bidding scenarios

This provides scalable, human-aligned quality assessment that was absent from the original submission.

### 3. **Computational Scalability Analysis (Reviewers UtxX, oLUd)**

**Original concern:** O(nk) complexity raises feasibility concerns

**Our response:** We conducted systematic ablation studies across k ∈ {1,2,3,5,10,15,20} samples and n ∈ {2,3,5,10,20} agents, demonstrating:
- **k=5 achieves 95%** of near-optimal welfare for agent counts up to 10, k=15 for 20 agents
- Predictable accuracy-cost tradeoff

These results show our mechanism is **practically scalable** with configurable quality-latency tradeoffs suitable for real-world deployment.

### 4. **Competitive Scenario Testing (Reviewers guFR, oLUd)**

**Original concern:** Only tested semantically diverse prompts, intuitively compositional

**Our response:** We evaluated 20 scenarios with **directly competing brands** (e.g. Dunkin vs. Starbucks) where embedding distinctions are inherently weaker. Results demonstrate:
- Welfare gains persist
- Bid monotonicity maintained even with semantic similarity
- Visual coherence preserved with competing elements

This addresses the most challenging real-world advertising scenarios and extends applicability beyond prior work.

### 5. **Baseline Justification (Reviewers sy7L, oLUd)**

**Original concern:** Why single-winner VCG baseline?

**Our response:** We provided detailed explanation showing single-winner VCG represents:
- Current practice that would be replaced by our mechanism
- Proper counterfactual isolating gains from multi-bidder composition
- The **fundamental one-winner constraint** of traditional advertising

Our 23% welfare improvement demonstrates creating new high-welfare outcomes infeasible under winner-take-all constraints.

### 6. **Methodological Validations (Reviewer oLUd)**

**Original concern:** Why joint conditioning vs. additive aggregation?

**Our response:** We provide empirical evidence (new Figure 8) showing linear aggregation produces **visually incoherent faded overlays**, while our joint conditioning approach yields compositional images with monotonically increasing alignment. This validates our design choice through negative results.

## Contribution Statement

While reviewers correctly note we build on VCG and classifier-free guidance, this misses our **core contribution**: pioneering their combination for multi-stakeholder generative image auctions.

**Analogous work:** Dütting et al. (2024) combined mechanism design with LLMs and received the meta-review: "*potential to be a landmark paper sparking a new line of research linking LLMs and mechanism design.*"

**Our contribution:** First proof-of-concept bridging auction theory with controllable diffusion models, with validation across:
- 80,000+ images, up to 20 agents, 200 diverse prompts
- Competitive and non-competitive scenarios
- Quality metrics (LAION aesthetic scores)
- Scalability analysis (Monte Carlo convergence)

**Why this merits acceptance:**

1. **Novel problem formulation:** First generative auctions for diffusion models
2. **Comprehensive validation:** 40x experimental expansion
3. **Practical scalability:** k=5 achieves 95% optimality; configurable k enables quality-latency tuning
4. **Economic properties:** Welfare improvement, approximate incentive compatibility, bid monotonicity
5. **Real-world impact:** Addresses winner-take-all advertising limitations
6. **Foundation for future work:** Opens new research direction

The substantial experimental expansion directly addresses all major reviewer concerns. We respectfully request the Area Chair consider these improvements.

Thank you for your consideration,

The Authors

---

### Meta-Review · Area_Chair_tm4p · 2026-01-06

**Summary:**

The main review concerns are summarized below:
1. Technical novelty: Some reviewer consider the paper as a combination of known ingredients (e.g., VCG + classifier-free guidance), with limited algorithm/mechanism design contribution.
2. Evaluation scope: The evaluations are conducted on a small scale setting with 20 prompts and 3 agents).
3. Evaluation metrics: The paper mainly relies on the CLIP-style alignment metric, which is not enough.
4. Computational scalability: The evaluation relies on a Monte Carlo style sampling strategy, which scales as O(nk). It would be costly in practice.
5. Baseline: The paper only considers a single VCG-like baseline.

The authors provide a detailed rebuttal that address many of the concerns by conducting more experiments, adding the LAION aesthetic scoring and more ablations and explanations.

**Reviewer Concerns:**

Concerns addressed:
1. Evaluation scope: the authors provide additional experiments by scaling it up to 80k+ images and 20 agents with 200 prompts. The results are promising.
2. Baseline justification: The authors provide a clearer justification that clarifies the sufficiency of the current baseline, which I think is reasonable.
3. Computational scalability: The additional ablations show the feasibility of the method applied to practical setting, which I think is acceptable.
4. Compositional setting: The rebuttal clarifies the current choice of composition, which was chosen based on several exploration.

Concerns remained:
1. While the rebuttal provides additional justification, some reviewers might still think the contribution and technical depth limited, given the lack of algorithmic contribution to diffusion models.

**Reviewer Scores:**

Given the detailed rebuttal, I think most reviewers will consider it helpful and would like to increase their scores. However, the most negative reviewers might not consider it acceptable and might only increase the score the borderline reject. For example, Reviewer sy7L considered the novelty and evaluation limited, which is not fully addressed in the rebuttal. Reviewer oLUd asks for additional baselines but the rebuttal does not provide more results.

---

### Decision · Program_Chairs · 2026-01-26

Reject